# Underestimation of Radiation Doses by Compliance of Wearing Dosimeters among Fluoroscopically-Guided Interventional Medical Workers in Korea

**DOI:** 10.3390/ijerph19148393

**Published:** 2022-07-09

**Authors:** Won Jin Lee, Eun Jin Jang, Kyeong Seo Kim, Ye Jin Bang

**Affiliations:** Department of Preventive Medicine, Korea University College of Medicine, Seoul 02841, Korea; zziny@korea.ac.kr (E.J.J.); knn5655@korea.ac.kr (K.S.K.); byj310@korea.ac.kr (Y.J.B.)

**Keywords:** badge, fluoroscopically-guided interventional procedures, hospital workers, occupational exposure, thermoluminescent dosimetry

## Abstract

This study aimed to estimate the level of underestimation of National Dose Registry (NDR) doses based on the workers’ dosimeter wearing compliance. In 2021, a nationwide survey of Korean medical radiation workers was conducted. A total of 989 medical workers who performed fluoroscopically-guided interventional procedures participated, and their NDR was compared with the adjusted doses by multiplying the correction factors based on the individual level of dosimeter compliance from the questionnaire. Ordinal logistic regression analysis was performed to identify the factors for low dosimeter wearing. Based on the data from the NDR, the average annual effective radiation dose was 0.95 mSv, while the compliance-adjusted dose was 1.79 mSv, yielding an 89% increase. The risks for low compliance with wearing a badge were significantly higher among doctors, professionals other than radiologists or cardiologists, workers not frequently involved in performing fluoroscopically-guided interventional procedures, and workers who did not frequently wear protective devices. This study provided quantitative information demonstrating that the NDR data may have underestimated the actual occupational radiation exposure. The underestimation of NDR doses may lead to biased risk estimates in epidemiological studies for radiation workers, and considerable attention on dosimetry wearing compliance is required to interpret and utilize NDR data.

## 1. Introduction

Data on the level of occupational radiation exposure in workers is periodically obtained from individual dosimeters worldwide and stored in the National Dose Registry (NDR). Such registries are important parts of national occupational radiation protection programs in many countries. The NDR data are essential to protect radiation workers from radiation exposures and facilitate the conduct of epidemiological studies that estimate the effects of low dose and low dose rate ionizing radiation [1,2]. Therefore, the accuracy of the reported dose data should be the essential premise of the usage dose registries. 

However, some radiation workers do not always wear dosimeters due to various reasons, such as reduction in their work performance and discomfort while working, and problems with regulatory authorities for exceeding the dose limits [3,4,5]. Lower dosimeter wearing compliance could cause inaccurate dose measurements, which leads to an underestimation of the doses received by workers. Therefore, the health risks reported in the epidemiologic studies would be biased when the doses are assessed based only on the NDR data.

Among radiation workers, the medical workers who perform fluoroscopically-guided interventional procedures are exposed to a substantially higher dose compared with that of other medical radiation workers performing conventional radiography [6]. Hence, an effective radiation protection program should be implemented to protect medical workers from radiation exposure during fluoroscopically-guided interventional procedures [7]. Some health effects from occupational radiation exposure have been reported among fluoroscopically-guided interventional medical workers [8]. Therefore, it is important that occupational radiation doses are accurately monitored to determine the effective radiation protection measures and to conduct epidemiological studies in this group.

This study aimed to investigate the level of potential underestimation of occupational radiation doses among medical workers who perform fluoroscopically-guided interventional procedures. Identifying the level of NDR underestimation and related factors may serve as a fundamental step in developing strategies to protect radiation workers against occupational radiation exposure and in applying the NDR badge doses in radiation epidemiology studies.

## 2. Materials and Methods

### 2.1. Study Population

A web-based nationwide survey was conducted from June 2021 to August 2021. The target population included all diagnostic medical radiation workers who performed or assisted in interventional fluoroscopy procedures and were registered at an NDR in the Korea Disease Control and Prevention Agency (KDCA), which operates a lifetime management system for occupational radiation doses. The registry includes physicians, dentists, dental hygienists, radiologic technologists, nurses, and medical assistants. 

The participants were invited through the nationwide branches of 15 professional societies that specialize in interventional radiology such as the Korean Society of Interventional Radiology, the Korean Society of Interventional Cardiology, the Korean Pain Interventional Society, the Korean Orthopedic Association, the Korean Society of Interventional Neuroradiology, and the Korean Pancreatobiliary Association. A total of 989 medical workers who reported wearing a TLD (thermoluminescent dosimeter) badge beneath their apron on the left side of the chest (standard protocol in South Korea) participated in this study. Written informed consent, with permission to use the radiation dosimeters, was voluntarily provided by all study participants prior to the study enrollment. This study was reviewed and approved by the Institutional Review Board of Korea University (KUIRB-2021-0181-01).

### 2.2. Questionnaire

A participant-friendly online self-administered questionnaire (http://www.rhs.kr/intervention, accessed on 20 June 2022) for interventional medical radiation workers was developed based on our previous study [9]. To maximize the participation rate, several approaches were applied, such as maintaining periodic contacts with the executives and publicity team of the relevant professional societies, asking them to link their website to the web survey, creating banner advertisements that promote the study on their websites, sending e-mails to introduce the sending web surveys to individual members, making reminder calls as follow-ups to invitations, providing gift vouchers to encourage participation, and sending a statement from the KDCA to confirm the participants’ official cooperation to the related societies. 

The questionnaires contained questions to assess the participants’ demographic characteristics and lifestyle (age, sex, name, smoking and alcohol consumption status), work history (job title, medical specialty, year of entering the field, duration of employment, and type of medical facility), work practices (name of procedures performed, number of days performing interventional fluoroscopy procedures per month, frequency of performing fluoroscopically-guided interventional procedures per week, and proportion of interventional procedures performed during their practice), and radiation safety practices (wearing of protective devices). To estimate the actual level of radiation exposure, the participants were asked how often they wear a personal monitoring badge during their practice (i.e., 100%, 75–99%, 25–74%, and <25%).

### 2.3. Dosimetry Data

The NDR of KDCA collects the information of registered diagnostic radiation workers who are required to wear a TLD badge mandatorily. The registry contains radiation dose records for 97,801 individuals being monitored in 2020 [10]. The dose data were reported quarterly by five centers that provide personal dosimetry service designated by the KDCA. The national dose registry contains the workers’ name, sex, date of birth, personal identification number, workplace address, job title, quarterly measured dose data, and the beginning and end of the period of measurement. The NDR data of the study participants were requested from the KDCA, and an informed consent for the use of personal information was obtained. Each participant’s information was linked to the NDR database by matching the name, sex, and date of birth. The doses were reported as personal dose equivalent at a depth of 10 mm [Hp (10)] by the reading system based on the standard method used by the NDR in accordance with the Regulations for Safety Management of Diagnostic Radiation [11]. The minimum detectable quarterly level of the NDR is 0.01 mSv. In cases where the dose was below the minimum detectable level, the dose was considered to be half of the detectable level due to a highly skewed distribution [12].

### 2.4. Data Analysis

An adjusted NDR recorded dose was calculated by multiplying the correction factor based on the compliance level of wearing badges. The correction factors were assigned scores of 1, 1.15, 2, and 8, respectively, based on the reciprocal of the participants’ responses to the question, “how often do you wear a badge during your practice?” (i.e., 1 for 100%, 1.15 for 75–99%, 2 for 25–74%, and 8 for <25%). The percentage increase in radiation dose after incorporating the compliance level was calculated using the following equation: [100 × (adjusted dose − NDR dose)/NDR dose]. An ordinal logistic regression analysis was used to examine the association between the compliance of dosimeter wearing and occupational characteristics after adjusting for age and sex. In this analysis, workers were categorized as four compliance groups of dosimeter wearing (i.e., 100%, 75–99%, 25–74%, and <25%) as the outcome variable. All statistical analyses were performed using the R software, version 4.1.1 (R Core Team, Vienna, Austria).

## 3. Results

Demographic and occupational characteristics of the study participants according to compliance of badge wearing are presented in Table 1. Among the total participants, 521 workers wore a badge regularly (52.7%). The majority of the workers were men (73.4%), younger than 50 years of age (81.2%), and worked in general hospitals (86.0%). More than half of the participants started working after 2010, and the average duration of experience was eight years (standard deviation, 6.4). The demographic and occupational characteristics were similar based on the compliance level of wearing a badge.

The average annual occupational radiation doses before and after adjustment by level of badge wearing compliance are summarized in Table 2. The average annual effective radiation dose based on the NDR data was 0.95 mSv, and the compliance-adjusted dose was 1.79 mSv, yielding an 89% increase from the official NDR report. The workers who wore the badge occasionally (<25%) received only half of the radiation doses (0.62 mSv) compared with the doses of workers who wore the badge always (1.10 mSv) based on the NDR data; however, the adjusted doses from the compliance of badge wearing were more than 4.5-fold higher (4.97 mSv) than those of workers wearing the badge always (1.10 mSv).

The results of ordinal logistic regression analyses with low badge wearing compliance after adjustment for age and sex by occupational characteristics are presented in Table 3. The risks for low badge wearing compliance were significantly higher among doctors [odds ratio (OR) = 1.59; 95% confidence interval (CI): 1.20–2.11) and professionals other than radiologists or cardiologists (OR = 1.57; 95% CI: 1.18–2.09). The workers who did not frequently perform fluoroscopically-guided procedures had increased risks for low badge wearing compliance compared with full-time medical workers who perform fluoroscopically-guided interventional procedures. The risks also increased among the workers who did not wear protective devices regularly (i.e., lead apron, thyroid shield, and goggle).

## 4. Discussion

This study indicated that the NDR data could underestimate the actual occupational radiation exposure level among medical workers who perform fluoroscopically-guided interventional procedures by approximately 90%, and a few subgroups had low dosimeter badge wearing compliance. These findings suggest that it is necessary to monitor the dose uncertainty when using the NDR dose in the epidemiologic studies to obtain more reliable estimates of the level of radiation exposure by conducting a sensitivity analysis excluding workers who had low reliable NDR doses. Therefore, it is important to increase the badge wearing compliance that can reflect the actual radiation exposure level and collect the data on the badge compliance proportion to validate the reported dose data. This study could add quantitative evidence for emphasizing the appropriate use of the NDR dose in epidemiologic studies and help develop workplace policies by increasing the dosimeter wearing compliance.

Our findings indicated that the actual occupational radiation doses could be higher among low compliance dosimeter wearers than among always wearers, although the official NDR data reported the highest values among always wearers as compared to other groups of dosimeter wearers (i.e., 75–99%, 25–74%, and <25%). The difference in the estimated actual doses among study participants was increased with decreasing the compliance level of dosimeter wearing. Therefore, the risk reported in epidemiologic studies would be biased when the doses are only assessed based on the reported NDR doses, especially among irregular badge wearers. The underestimated doses caused by censoring of minimum detection limit doses resulted in the overestimation of the radiation exposure risk in the Canadian occupational radiation workers [13], while the risk for all-cause mortality in the United States (US) Oak Ridge National Laboratory radiation workers was not significantly altered after taking into consideration the random measurement error and missed doses due to falling below the minimum detection level [14]. The estimates of external radiation dose obtained from personal dosimeters have several quantifying sources of errors [15]; therefore, its influence on risk estimation could vary at different levels of exposure underestimation and needs to be evaluated in each epidemiological study.

The level of underestimation reported in the present study (approximately 90%) may be comparable to that among Korean interventional radiologists [9] when considering the lower badge wearing compliance among doctors compared with that in other medical workers. This previous study also reported that the validity of NDR was significantly dependent on the badge wearing compliance level. However, the findings should be interpreted cautiously, as the reported NDR doses may not entirely depend on the level of dosimeter badge wearing compliance. The dosimeter dose has been known to have other uncertainty factors which may influence the NDR doses, such as inappropriate use of the dosimeter, damage to the dosimeter during use or processing, and faulty conditions of the evaluating equipment [16]. In addition, a single dosimeter badge may not be sufficient to measure the radiation exposure dose to all parts of the body, and omits the exposure dose delivered to the unprotected body; therefore, the use of two monitoring badges was recommended to measure the occupational dose among interventional radiology staffs accurately [17], although our study participants wore one dosimeter. Considering all these potential uncertainties, the actual radiation exposure doses in our medical workers could be higher than our estimation, which was adjusted by a single source of error.

Our findings on a few subgroups who had significantly higher risks of wearing a badge irregularly (doctors, professionals other than radiologists or cardiologists, etc.) may be attributable to their occupational characteristics. Doctors may feel uncomfortable wearing the badge while working and are more sensitive when prohibited from being exposed to radiation that exceed the dose limits compared with the radiologic technologists and nurses, thus increasing the tendency to be less compliant in wearing a badge. Workers who specialize in radiology and cardiology may have more radiation safety education than in other specialties because they are more frequently involved in the performance of fluoroscopically-guided interventional procedures. Radiologists have received more training and more knowledge on radiation exposure compared with doctors of other medical specialties in Spain [18] and the US [19], and radiologists reported a higher accuracy of radiation doses associated with a standard chest x-ray exposure than non-radiologists in Hong Kong [20]. However, orthopedic surgeons less frequently wore a dosimeter (29.2%) compared with doctors with other specialties and did not apply the standard rules for radiation safety in South Korea [21]. Orthopedic surgeons in Ireland (15%) [22] and worldwide (approximately 25%) [23], pathologists in Australia (36%) [24], and urologists in the US (35%) [25] have lower compliance with badge wearing compared with that reported in our study, therefore, the level of radiation dose underestimation would be higher in these populations than that reported in this study.

In addition, workers who less frequently perform fluoroscopically-guided interventional procedures are not informed about the need to wear a badge since their job does not always involve exposure to radiation. It is also possible that workers who wear dosimeters are more cautious about performing fluoroscopy [25], and this may lead to the increased frequency of wearing personal protective devices. The greater tendency to wear a badge and personal protective devices among workers who frequently performed fluoroscopically-guided interventional procedures compared with those who less frequently performed fluoroscopically-guided interventional procedures in this study support this interpretation.

However, the validity of NDR data could vary depending on the characteristics of the study participants. For example, no difference was found in the compliance level between sexes, positions, and hospital types and sizes among radiologists in Jordan [26]. Therefore, further studies are warranted to identify the risk factors for the compliance to wearing a dosimeter, and the reliable subgroups of NDR dose should be examined; moreover, only specific populations who have a high compliance level could be selected when the NDR data are used in epidemiologic studies.

The proportion of regular dosimeter wearers in our study (53%) was generally comparable with that in previous studies conducted in South Korean radiologic technologists (66%) [27], US cardiologists (52%) [28], and radiographers in South Africa (67.2%) [29], but lower than those of radiologists in Jordan (93.5%) [26] and general surgeons in the US (84%) [30]. Since our study participants were doctors, nurses, and radiologic technologists who are involved in fluoroscopy-guided interventional procedures, more intensive intervention efforts to improve the compliance of badge wearing and experimental study applying the intervention are needed in the workplace.

Our study has some limitations. The questionnaire related to dosimeter wearing compliance represents the overall compliance level of radiation workers and may not capture detailed information about the frequency of wearing a badge by time period and specific work procedures. In addition, the correction factor used in this study was based on a self-assessment in the questionnaire. However, the compliance level may be non-differentially misclassified among the participants, and information on self-reported working practices that involve radiation exposure has been generally reported as reliable among South Korean radiologic technologists [31]. Further research with information on time period and work procedures may help clarify the specific role of badge-wearing compliance. The other limitation is the small sample size, which might limit the generalizability of the findings. However, the survey included various nationwide branches of professional societies that specialize in fluoroscopically-guided interventional procedures and assumed that workers in these fields represent the available working interventional medical workers in South Korea [32].

## 5. Conclusions

In summary, we provided quantitative information demonstrating that the NDR data may have underestimated the actual occupational radiation exposure level among medical radiation workers who perform fluoroscopically-guided interventional procedures in South Korea. Such an underestimation of occupational radiation doses may lead to biased risk estimates in epidemiological studies of radiation workers. Thus, considerable attention is required when interpreting and utilizing the NDR data, such as conducting a sensitivity analysis excluding workers who had low reliable NDR doses. Further studies are needed to evaluate the influence of the dose uncertainty induced by the underestimation of the true doses due to the low dosimeter wearing compliance.

## Figures and Tables

**Table 1 ijerph-19-08393-t001:** Demographic and occupational characteristics of the study participants by level of dosimeter wearing compliance.

Characteristics ^1^	Total		Compliance of Badge Wearing							
			100%		75–99%		25–74%		<25%	
	*N*	% ^2^	*N*	% ^3^	*N*	% ^3^	*N*	% ^3^	*N*	% ^3^
Total	989	100.0	521	52.7	167	16.9	136	13.8	165	16.7
Sex										
Male	726	73.4	374	51.5	129	17.8	97	13.4	126	17.4
Female	263	26.6	147	55.9	38	14.4	39	14.8	39	14.8
Age (years)										
<30	73	7.4	41	56.2	14	19.2	7	9.6	11	15.1
30–34	153	15.5	83	54.2	25	16.3	16	10.5	29	19.0
35–39	214	21.6	103	48.1	41	19.2	30	14.0	40	18.7
40–44	238	24.1	125	52.5	43	18.1	36	15.1	34	14.3
45–49	125	12.6	73	58.4	18	14.4	15	12.0	19	15.2
≥50	186	18.8	96	51.6	26	14.0	32	17.2	32	17.2
Type of medical facility										
General hospitals	851	86.0	439	51.6	140	16.5	121	14.2	151	17.7
Others	138	14.0	82	59.4	27	19.6	15	10.9	14	10.1
Occupation										
Technologists	469	47.4	271	57.8	71	15.1	52	11.1	75	16.0
Doctors	274	27.7	122	44.5	52	19.0	53	19.3	47	17.2
Nurses	246	24.9	128	52.0	44	17.9	31	12.6	43	17.5
Specialty										
Radiology	558	56.4	317	56.8	92	16.5	59	10.6	90	16.1
Cardiology	207	20.9	108	52.2	27	13.0	37	17.9	35	16.9
Others	224	22.6	96	42.9	48	21.4	40	17.9	40	17.9
Calendar year of beginning work										
<2010	326	33.0	179	54.9	48	14.7	48	14.7	51	15.6
2010–2014	233	23.6	119	51.1	35	15.0	39	16.7	40	17.2
≥2015	430	43.5	223	51.9	84	19.5	49	11.4	74	17.2
Duration of employment (years)										
<5	369	37.3	193	52.3	77	20.9	38	10.3	61	16.5
5–9	275	27.8	152	55.3	38	13.8	44	16.0	41	14.9
≥10	345	34.9	176	51.0	52	15.1	54	15.7	63	18.3
Number of days performing interventional fluoroscopy per month										
<10	132	13.3	66	50.0	24	18.2	18	13.6	24	18.2
10–19	178	18.0	90	50.6	29	16.3	31	17.4	28	15.7
≥20	567	57.3	307	54.1	89	15.7	74	13.1	97	17.1
Frequency of performing interventional fluoroscopy per week										
<10	245	24.8	118	48.2	42	17.1	41	16.7	44	18.0
10–29	279	28.2	138	49.5	52	18.6	44	15.8	45	16.1
≥30	346	35.0	202	58.4	49	14.2	36	10.4	59	17.1
Proportion of interventional fluoroscopy procedures performed during their practice										
100%	296	29.9	181	61.1	37	12.5	28	9.5	50	16.9
75–99%	199	20.1	101	50.8	41	20.6	25	12.6	32	16.1
50–74%	146	14.8	78	53.4	20	13.7	21	14.4	27	18.5
25–49%	125	12.6	51	40.8	22	17.6	35	28.0	17	13.6
<25%	112	11.3	53	47.3	22	19.6	15	13.4	22	19.6
Wearing a lead apron										
Always	926	93.6	500	54.0	145	15.7	128	13.8	153	16.5
Not always	63	6.4	21	33.3	22	34.9	8	12.7	12	19.0
Wearing a thyroid shield										
Always	818	82.7	447	54.6	124	15.2	112	13.7	135	16.5
Not always	171	17.3	74	43.3	43	25.1	24	14.0	30	17.5
Wearing a goggle shield										
Always	215	21.7	132	61.4	30	14.0	25	11.6	28	13.0
Not always	774	78.3	389	50.3	137	17.7	111	14.3	137	17.7
Wearing a glove shield										
Ever	146	14.8	81	55.5	33	22.6	18	12.3	14	9.6
Never	843	85.2	440	52.2	134	15.9	118	14.0	151	17.9

^1^ Numbers may not add up to the total due to missing information. ^2^ Column percent. ^3^ Row percent.

**Table 2 ijerph-19-08393-t002:** Annual occupational radiation doses before and after complying with dosimeter wearing.

Dosimeter Wearing Compliance	Number	NDR (mSv) (Mean ± SD)	Adjusted Dose (mSv) (Mean ± SD)	Difference (mSv) (Mean ± SD)	% Increases
Total	989	0.95 ± 1.33	1.79 ± 3.34	0.85 ± 2.83	89.4
100%	521	1.10 ± 1.45	1.10 ± 1.45	0.00 ± 0.00	0.0
75–99%	167	0.99 ± 1.48	1.14 ± 1.70	0.15 ± 0.22	15.0
25–74%	136	0.70 ± 1.00	1.41 ± 1.99	0.70 ± 1.00	100.0
<25%	165	0.62 ± 0.81	4.97 ± 6.50	4.35 ± 5.68	700.0

**Table 3 ijerph-19-08393-t003:** Ordinal logistic regression analyses of the low dosimeter wearing compliance by occupational characteristics.

	Crude OR (95% CI)	Adjusted OR (95% CI) ^1^
Sex		
Female	1.00 (ref)	1.00 (ref)
Male	1.16 (0.89–1.52)	1.15 (0.87–1.51)
Age (years)		
<30	1.00 (ref)	1.00 (ref)
30–34	1.15 (0.68–1.98)	1.14 (0.67–1.95)
35–39	1.38 (0.84–2.31)	1.36 (0.82–2.27)
40–44	1.15 (0.70–1.90)	1.12 (0.68–1.87)
45–49	0.97 (0.56–1.70)	0.95 (0.54–1.66)
≥50	1.28 (0.77–2.16)	1.23 (0.73–2.09)
Type of medical facility		
General hospitals	1.00 (ref)	1.00 (ref)
Others	0.67 (0.47–0.95)	0.66 (0.46–0.94)
Occupation		
Technologists	1.00 (ref)	1.00 (ref)
Doctors	1.55 (1.17–2.04)	1.59 (1.20–2.11)
Nurses	1.22 (0.91–1.64)	1.44 (1.01–2.06)
Specialty		
Radiology	1.00 (ref)	1.00 (ref)
Cardiology	1.24 (0.92–1.68)	1.25 (0.92–1.69)
Others	1.56 (1.17–2.08)	1.57 (1.18–2.09)
Calendar year of beginning work		
<2010	0.93 (0.71–1.22)	0.85 (0.59–1.21)
2010–2014	1.08 (0.80–1.46)	1.02 (0.75–1.40)
≥2015	1.00 (ref)	1.00 (ref)
Duration of employment (years)		
<5	1.00 (ref)	1.00 (ref)
5–9	0.95 (0.71–1.28)	0.96 (0.71–1.29)
≥10	1.14 (0.86–1.50)	1.17 (0.84–1.63)
Number of days performing interventional fluoroscopy per month		
<10	1.00 (ref)	1.00 (ref)
10–19	0.97 (0.64–1.48)	0.97 (0.64–1.48)
≥20	0.88 (0.62–1.25)	0.87 (0.60–1.25)
Frequency of performing interventional fluoroscopy per week		
<10	1.00 (ref)	1.00 (ref)
10–29	0.92 (0.67–1.27)	0.92 (0.67–1.27)
≥30	0.71 (0.52–0.97)	0.71 (0.52–0.97)
Proportion of interventional fluoroscopy performed during their practice		
100%	1.00 (ref)	1.00 (ref)
75–99%	1.34 (0.95–1.89)	1.36 (0.96–1.92)
50–74%	1.35 (0.92–1.98)	1.37 (0.93–2.02)
25–49%	1.85 (1.26–2.72)	1.87 (1.26–2.76)
<25%	1.58 (1.05–2.39)	1.63 (1.06–2.50)
Wearing a lead apron		
Always	1.00 (ref)	1.00 (ref)
Not always	1.62 (1.04–2.51)	1.63 (1.05–2.54)
Wearing a thyroid shield		
Always	1.00 (ref)	1.00 (ref)
Not always	1.34 (0.99–1.80)	1.34 (0.99–1.81)
Wearing a goggle shield		
Always	1.00 (ref)	1.00 (ref)
Not always	1.53 (1.14–2.07)	1.57 (1.17–2.13)
Wearing a glove shield		
Ever	1.00 (ref)	1.00 (ref)
Never	1.31 (0.95–1.84)	1.35 (0.97–1.90)

^1^ Adjusted for sex and age (continuous). OR, Odds ratio. CI, Confidence interval.

## Data Availability

Access to detailed individual data is restricted for both legal and ethical concerns. The availability of data should be reviewed with the corresponding author and approved by the Institutional Review Board of Korea University.

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
