# Peer review of "Underestimation of Radiation Doses by Compliance of Wearing Dosimeters among Fluoroscopically-Guided Interventional Medical Workers in Korea"

_ijerph, 2022, doi:10.3390/ijerph19148393_

Round 1
Reviewer 1 Report
Underestimation of radiation doses and compliance with wearing dosimeters among medical workers who perform FGI is actually presenting the survey results among FGI professionals showing that there are people that do not wear dosimeters regularly. This is important, but has been known already (especially in hospital environment) and can be found in literature. The simple correction factor is introduced and this is different from other similar studies. Since, the correction factor is based on a self-assessment, a number of uncertainties are connected to it and this has not been investigated. The scientific soundness of the paper would be improved if this has been done.
Author Response
Response: We thank the reviewer for their helpful comments and hope that our revisions meet the requirements of the reviewer. We have included the page and line numbers in parentheses, referring to the revised file with the track changes on.
To emphasize the originality of our study, we have added the sentence “This study provided quantitative information of demonstrating that the NDR data may have underestimated the actual occupational radiation exposure” in the Abstract (page 1, line 21-22).
To incorporate the reviewer’s comment on the limitation of the correction factor, we have added the sentence “In addition, the correction factor used in this study was based on a self-assessment in the questionnaire” in the Discussion (page 9, line 273-274).
Reviewer 2 Report
The manuscript is clearly improved by the revision that has taken place. Some points should still be reconsidered:
1. (line 14f) Ordinal regression is the method that was applied to identify potentially prognostic variables. It would be good to mention this in the abstract.
2. (line 151f) When examining your data with ordinal regression, it is not appropriate to dichotomize the data in a first step – the ordinal regression is the appropriate method for four categories in the outcome variable (after verification of the model requirements). Dichotomization would result in a binary logistic regression. Clarify whether this is an ordinal regression or a binary logistic regression. The results have to fit the clearly defined and applied method.
3. (line 189ff and Table 3) Another follow-up question to the point already made: are these results the results of an ordinal regression with four categories in the target variable or the results of a binary logistic regression. The wording and presentation of the results slightly suggest binary logistic regression. Both methods are applicable to the research question, but the method used should be clearly stated.
Author Response
Response: We thank the reviewer for their helpful comments and hope that our revisions meet the requirements of the reviewer. We have included the page and line numbers in parentheses, referring to the revised file with the track changes on.
According to the reviewer’s previous comments, we have conducted ordinal regression logistic analysis based on four categories in the outcome variable (i.e., 100%, 75%–99%, 25%–74%, and < 25% of dosimeter wearing compliance) and replaced Table 3 with the findings from the ordinal logistic regression analysis. As the reviewer pointed, however, there a few wordings that we should revise to clarify.
To avoid any confusion, we have revised the sentences from “Ordinal logistic regression analysis was used to identify the risk factors for not always wearing dosimeter after adjusting for age and sex. In this analysis, workers were categorized as always and not always badge wearers” to “Ordinal logistic regression analysis was used to examine the association between the compliance of dosimeter wearing and occupational characteristics after adjusting for age and sex. In this analysis, workers were categorized as four compliance groups of dosimeter wearing (i.e., 100%, 75%–99%, 25%–74%, and < 25%) as the outcome variable” in the Data analysis (page 3, line 126-131).
We have also revised the sentence as “Ordinal logistic regression analysis was performed to identify the risk factors for low dosimeter wearing” in the Abstract (page 1, line 14-15).
To avoid ambiguity between ordinal regression and binary logistic regression. we have also revised the sentence from “Our findings indicated that the actual occupational radiation doses could be higher among not always dosimeter wearers than among always wearers although the official NDR data reported higher values among always wearers than among not always wearers. The difference in the estimated actual doses between always and not always badge wearers was larger based on the level dosimeter wearing compliance. Therefore, the risk reported in epidemiologic studies would be biased when the doses are only assessed based on the reported NDR doses, especially among not always badge wearers” to “Our findings indicated that the actual occupational radiation doses could be higher among low compliance dosimeter wearers than among always wearers although the official NDR data reported highest values among always wearers than among other groups of dosimeter wearers (i.e., 75%–99%, 25%–74%, and < 25%). The difference in the estimated actual doses among study participants was increased with decreasing the compliance level of dosimeter wearing. Therefore, the risk reported in epidemiologic studies would be biased when the doses are only assessed based on the reported NDR doses, especially among irregular badge wearers” in the Discussion (page 7, line 191- page 8, line 199).
Round 2
Reviewer 1 Report
Accept
This manuscript is a resubmission of an earlier submission. The following is a list of the peer review reports and author responses from that submission.
Round 1
Reviewer 1 Report
The paper under the title: Underestimation of radiation doses and compliance with wearing dosimeters among interventional medical workers is actually presenting the survey results among FGI professionals showing that there are people that do not wear dosimeters regularly. This has been known already and can be found in literature, but the data are on Korean population what can be of interest for comparison with other surveys.
Nevertheless, the title does not reveal this and should be changed.
It will be more interesting if the study is repeated after some actions to make people wear dosimeters more regularly? Then it could be more convincing.
The abstract should be rewritten. It is hard to understand what have been done. E.g. correction factors are derived how? Then, the OR is given for two categories, but not for other two (why?).
The term 'interventional medical radiation workers' is unusual and misleading. Please use e.g: medical workers who perform fluoroscopic-guided interventional procedures (as already used in the text)
In interventional radiology (and all FGI), for an accurate determination of effective dose to the staff, measurements with two dosemeters have been recommended (ICRP).
Author Response
The paper under the title: Underestimation of radiation doses and compliance with wearing dosimeters among interventional medical workers is actually presenting the survey results among FGI professionals showing that there are people that do not wear dosimeters regularly. This has been known already and can be found in literature, but the data are on Korean population what can be of interest for comparison with other surveys.
We thank the reviewer for the helpful comments. We have revised the manuscript in a point-by-point manner according to your comments, as described below.
Nevertheless, the title does not reveal this and should be changed.
Response: As the reviewer pointed out, there are some studies on the compliance of badge wearing among workers; however, the direct estimation of dose underestimation by compliance with badge wearing has not been considered so far. Therefore, the main purpose of this study was to investigate the quantitative estimates of radiation dose underestimation among workers rather than to investigate compliance with badge wearing. This study adds quantitative evidence to emphasize the appropriate use of recorded NDR doses in epidemiologic studies. We believe that our quantitative estimates of radiation dose underestimation is the first to be reported among medical radiation workers as well as the first to be conducted in Korea.
To clarify these points, we have revised the title from “Underestimation of radiation doses and compliance with wearing dosimeters among interventional medical workers” to “Underestimation of radiation doses by compliance of wearing dosimeters among fluoroscopically-guided interventional medical workers in Korea.”
It will be more interesting if the study is repeated after some actions to make people wear dosimeters more regularly? Then it could be more convincing.
Response: We agree with the reviewer that the study before and after some actions to make people wear dosimeters more regularly could provide more convincing evidence. However, such a study should be designed as an experimental research, which is beyond the scope of this study. Our study was based on an observational design that aimed to investigate the level of potential underestimation of occupational radiation doses among medical workers who performed fluoroscopically guided interventional procedures using the reported TLD doses and questionnaire information.
We have added this point as a future research direction in the Discussion to incorporate the reviewer’s comment as “… more intensive intervention efforts to improve the compliance of badge wearing and experimental study applying the intervention are needed in the workplace” (page 9, line 268).
The abstract should be rewritten. It is hard to understand what have been done. E.g. correction factors are derived how? Then, the OR is given for two categories, but not for other two (why?).
Response: To follow the word limits of the Journal (200 words), we have provided the ORs for two categories but not for the other two in the Abstract. As the reviewer suggested, we have revised the Abstract by adding the method and keeping the consistency within the limit of the Journal’s guidelines as, “This study aimed to estimate the level of underestimation of National Dose Registry (NDR) doses based on the workers’ dosimeter wearing compliance. In 2021, a nationwide survey of Korean interventional medical radiation workers was conducted. A total of 989 medical workers who performed fluoroscopically guided interventional procedures participated, and their NDR was compared with the adjusted doses by multiplying the correction factors based on the individual level of dosimeter compliance from the questionnaire. Ordinal logistic regression analysis was performed to identify workers who had low dosimeter compliance. Based on the data from the NDR, the average annual effective radiation dose was 0.95 mSv, while the compliance-adjusted dose was 1.79 mSv, yielding an 89% increase. The risks for low compliance with wearing a badge were significantly higher among doctors, professionals other than radiologists or cardiologists, workers not frequently involved in performing fluoroscopically-guided interventional procedures, and workers who did not frequently wear protective devices. The underestimation of NDR doses may lead to biased risk estimates in epidemiological studies for radiation workers, and considerable attention on dosimetry wearing compliance is required to interpret and utilize NDR data.”
The term 'interventional medical radiation workers' is unusual and misleading. Please use e.g: medical workers who perform fluoroscopic-guided interventional procedures (as already used in the text)
Response: Thank you for this comment. We have replaced the term “interventional medical radiation workers” with “medical workers who perform fluoroscopically-guided interventional procedures” throughout the text, as well as the title based on the international literature.
In interventional radiology (and all FGI), for an accurate determination of effective dose to the staff, measurements with two dosemeters have been recommended (ICRP).
Response: We agree that a single dosimeter badge may not be sufficient to measure the radiation exposure dose to all parts of the body and omit the exposure dose delivered to the unprotected body; therefore, the use of two monitoring badges was recommended to accurately measure the occupational dose among interventional radiology staff. However, the current standard practice in South Korea is to provide a single dosimeter for medical radiation workers.
For clarity, we have revised the sentence in the Discussion as “In addition, a single dosimeter badge may not be sufficient to measure the radiation exposure dose to all parts of the body and omits the exposure dose delivered to the unprotected body; therefore, the use of two monitoring badges was recommended to measure the occupational dose among interventional radiology staffs accurately [17] although our study participants wore one dosimeter (page 8, line 217).
Reviewer 2 Report
Summary:
The manuscript aimes to report the self-reported wearing behaviour of badges for measuring radiation exposure of 989 people engaged in clinical environment and registered in the database NDR. First, the self-reported carrying behaviour is reported and secondly extrapolated values, corrected by the multiplication with the simple reciprocal of the rate of self-reported carrying the badge are reported. Logistic regression models are used to investigate irregular badge wearing for prognostic factors; estimators with adjustment for sex and age are additionally provided.
Some suggestions, ideas and questions:
1. Introduction (line 50ff)
The last sentence of the introduction suggests an unachievable goal in terms of "scientific evidence": we will get a better knowledge of the blur. This sentence should be reconsidered.
2. Section 2.3 (line 107f)
2a. Is the statement "The minimum detectable quarterly level of the NDR is 0.01mSv." right or is it rather "...the smallest reported unit in the NDR database is 0.01 mSv."
2b. An idea of the scope, i.e. the current data stock of the NDR database, would be interesting - just an approximate order of magnitude would be helpful.
2.4. Data analysis
If an equal distribution within the range of corridors is assumed, this should be explicitly stated. Additionally the correction factors should be explained as the reciprocal of the assumed values (as the mean of the margins).
Based on this assumption the second correction factor of 1.15 is not correct (first mentioned in line 112): The correction factor should be 1.13 based on 0.75 and 0.99. To stay with common percentages, a factor of 1.125 is also plausible for the limits 1.00 and 0.75, i.e. the mean value of 0.875, is assumed; Please explain your approach.
- line 116ff: Logistic regression was performed to detect irregular, i. e. not wearing badges in a perfect manner vs. regular, i. e. wearing badges perfect (100%). An ordinal regression should actually have a higher power and could be able to work out the results better. Have you considered ordinal regression?
3. Results
Table 2: Based on your correction factor of 1.15 the third increase in the table should be 15%.
Table 3: except two estimated effects "Occupation (Nurse)" and "Number of days performing interventional fluoroscopy per month (>=20)" there is no relevant difference between the crude and the adjusted estimations - what is the derivation here?
Table 3: at least one relation sign is not correct - see: "Calendar year of beginning work" - the year 2015.
4. Discussion
- line 192: The underestimation is about 47% (not 90% as you report) if your derived corrected value is right.
5. Conclusion (line 263ff)
Under-reporting and under-estimation leads to biased risk estimates and should be avoided. Is there a viable way to take this into account?
Author Response
Summary: The manuscript aimes to report the self-reported wearing behaviour of badges for measuring radiation exposure of 989 people engaged in clinical environment and registered in the database NDR. First, the self-reported carrying behaviour is reported and secondly extrapolated values, corrected by the multiplication with the simple reciprocal of the rate of self-reported carrying the badge are reported. Logistic regression models are used to investigate irregular badge wearing for prognostic factors; estimators with adjustment for sex and age are additionally provided.
We thank the reviewer for the helpful comments. We have revised the manuscript in a point-by-point manner according to your comments, as described below.
Some suggestions, ideas and questions:
- Introduction (line 50ff)
The last sentence of the introduction suggests an unachievable goal in terms of "scientific evidence": we will get a better knowledge of the blur. This sentence should be reconsidered.
Response: To avoid any ambiguity, as the reviewer pointed out, we have changed the sentence from “Identifying the level of NDR underestimation and related factors could provide scientific evidence that may serve as a fundamental step in developing strategies to protect radiation workers against occupational radiation exposure and in applying the NDR badge doses in radiation epidemiology studies” to “Identifying the level of NDR underestimation and related factors may serve as a fundamental step in developing strategies to protect radiation workers against occupational radiation exposure and in applying the NDR badge doses in radiation epidemiology studies” (page 2, line 56).
- Section 2.3 (line 107f)
2a. Is the statement "The minimum detectable quarterly level of the NDR is 0.01mSv" right or is it rather "...the smallest reported unit in the NDR database is 0.01 mSv."
Response: The sentence means that Quarterly doses below 0.01 mSv are the lowest detectable level of NDR. All dose data were based on measurements from thermoluminescent dosimeters and were calibrated annually.
2b. An idea of the scope, i.e. the current data stock of the NDR database, would be interesting - just an approximate order of magnitude would be helpful.
Response: The national dose registry in South Korea started in 1996 and maintains the records of occupational doses of ionizing radiation of diagnostic radiation workers. Currently, the registry contains the radiation dose records of 97,801 individuals currently being monitored (Korea Disease Control and Prevention Agency (KDCA), 2021).
As the reviewer suggested, we added the sentence “The registry contains radiation dose records for 97,801 individuals being monitored in 2020[10].” (page 3, line 104).
Korea Disease Control and Prevention Agency (KDCA). 2020 report on occupational radiation exposure in diagnostic radiology in Korea. 2021.
2.4. Data analysis
If an equal distribution within the range of corridors is assumed, this should be explicitly stated. Additionally the correction factors should be explained as the reciprocal of the assumed values (as the mean of the margins).
Response: Thank you for this comment. We have revised the sentence as ‘The correction factors were assigned scores of 1, 1.15, 2, and 8, respectively, based on the reciprocal of the participants’ responses to the question, “How often do you wear a badge during your practice?” (i.e., 1 for 100%, 1.15 for 75%–99%, 2 for 25%–74%, and 8 for < 25%)’ as the reviewer suggested (page 3, line 121).
Based on this assumption the second correction factor of 1.15 is not correct (first mentioned in line 112): The correction factor should be 1.13 based on 0.75 and 0.99. To stay with common percentages, a factor of 1.125 is also plausible for the limits 1.00 and 0.75, i.e. the mean value of 0.875, is assumed; Please explain your approach.
Response: We respectfully disagree with this comment. The reviewer suggested that the correction factor can be obtained as 1 + (1–0.87 or 1–0.875) = 1.13 or 1.125. However, this method is not appropriate for calculating the correction factor. For example, if we follow this calculation, the 25–74% group produces a correction factor of 1.5 (1 + [1-0.5]), which yields a 50% increase in the NDR badge dose. However, this group should have a 100% increase (i.e., 2.0, not 1.5) because only half wear badge. The same was true for other groups. Therefore, we apply the correction factor by 1/0.87, which gives 1.15 (as the reciprocal of the assumed values as the reviewer correctly expressed), not by 1–0.87. Using this approach, the NDR doses can be adjusted correctly; for example, the 25–74% group now has a correction factor of 2.0 (1/0.5). This is also the same approach we reported the validity of NDR doses among interventional radiologists previously (Ko et al., Occupational radiation exposure and validity of national dosimetry registry among Korean interventional radiologists. Int J Environ Res Public Health. 2021).
- line 116ff: Logistic regression was performed to detect irregular, i. e. not wearing badges in a perfect manner vs. regular, i. e. wearing badges perfect (100%). An ordinal regression should actually have a higher power and could be able to work out the results better. Have you considered ordinal regression?
Response: Thank you for your comment. For simplicity, we conducted dichotomous logistic regression because the distribution of the number of workers was similar between the always and not always badge-wearing groups. However, we agree that an ordinal regression analysis is more appropriate when the outcome variable is ordinal.
To address the reviewer’s suggestion, we conducted an ordinal regression analysis (please see the table below). The results were similar to those of dichotomous logistic regression, but they have the advantage of statistical power, as the reviewer pointed out. Therefore, we have replaced Table 3 with the findings from the ordinal logistic regression analysis.
Table. Binary logistic and ordinal logistic regression analyses of the dosimeter wearing compliance by occupational characteristics
|
Characteristics |
OR (95% CI)1 |
|
|
Binary logistic regression |
Ordinal logistic regression |
|
|
Sex |
|
|
|
Female |
1.00 (ref) |
1.00 (ref) |
|
Male |
1.20 (0.90–1.59) |
1.15 (0.87–1.51) |
|
Age (years) |
|
|
|
< 30 |
1.00 (ref) |
1.00 (ref) |
|
30–34 |
1.06 (0.61–1.87) |
1.14 (0.67–1.95) |
|
35–39 |
1.35 (0.79–2.31) |
1.36 (0.82–2.27) |
|
40–44 |
1.13 (0.67–1.93) |
1.12 (0.68–1.87) |
|
45–49 |
0.88 (0.49–1.59) |
0.95 (0.54–1.66 |
|
≥ 50 |
1.15 (0.66–2.00) |
1.23 (0.73–2.09) |
|
Type of medical facility |
|
|
|
General hospitals |
1.00 (ref) |
1.00 (ref) |
|
Others |
0.72 (0.50–1.04) |
0.66 (0.46–0.94) |
|
Occupation |
|
|
|
Technologists |
1.00 (ref) |
1.00 (ref) |
|
Doctors |
1.77 (1.31–2.41) |
1.59 (1.20–2.11) |
|
Nurses |
1.56 (1.06–2.30) |
1.44 (1.01–2.06) |
|
Specialty |
|
|
|
Radiology |
1.00 (ref) |
1.00 (ref) |
|
Cardiology |
1.21 (0.88–1.67) |
1.25 (0.92–1.69) |
|
Others |
1.77 (1.29–2.43) |
1.57 (1.18–2.09) |
|
Calendar year of beginning work |
|
|
|
< 2010 |
0.79 (0.54–1.15) |
0.85 (0.59–1.21) |
|
2010–2014 |
0.96 (0.69–1.35) |
1.02 (0.75–1.40) |
|
≥ 2015 |
1.00 (ref) |
1.00 (ref) |
|
Duration of employment (years) |
||
|
< 5 |
1.00 (ref) |
1.00 (ref) |
|
5–9 |
0.88 (0.64–1.21) |
0.96 (0.71–1.29) |
|
≥ 10 |
1.06 (0.74–1.51) |
1.17 (0.84–1.63) |
|
Number of days performing interventional fluoroscopy per month |
||
|
< 10 |
1.00 (ref) |
1.00 (ref) |
|
10–19 |
0.97 (0.62–1.53) |
0.97 (0.64–1.48) |
|
≥ 20 |
0.83 (0.56–1.22) |
0.87 (0.60–1.25) |
|
Frequency of performing interventional fluoroscopy per week |
||
|
< 10 |
1.00 (ref) |
1.00 (ref) |
|
10–29 |
0.94 (0.67–1.34) |
0.92 (0.67–1.27) |
|
≥ 30 |
0.66 (0.47–0.92) |
0.71 (0.52–0.97) |
|
Proportion of interventional fluoroscopy performed during their practice |
||
|
100% |
1.00 (ref) |
1.00 (ref) |
|
75%–99% |
1.56 (1.08–2.24) |
1.36 (0.96–1.92) |
|
50%–74% |
1.41 (0.94–2.11) |
1.37 (0.93–2.02) |
|
25%–49% |
2.34 (1.52–3.63) |
1.87 (1.26–2.76) |
|
< 25% |
1.84 (1.17–2.92) |
1.63 (1.06–2.50) |
|
Wearing a lead apron |
|
|
|
Always |
1.00 (ref) |
1.00 (ref) |
|
Not always |
2.37 (1.40–4.15) |
1.63 (1.05–2.54) |
|
Wearing a thyroid shield |
|
|
|
Always |
1.00 (ref) |
1.00 (ref) |
|
Not always |
1.59 (1.14–2.23) |
1.34 (0.99–1.81) |
|
Wearing a goggle shield |
|
|
|
Always |
1.00 (ref) |
1.00 (ref) |
|
Not always |
1.62 (1.19–2.22) |
1.57 (1.17–2.13) |
|
Wearing a glove shield |
|
|
|
Ever |
1.00 (ref) |
1.00 (ref) |
|
Never |
1.16 (0.81–1.66) |
1.35 (0.97–1.90) |
1Adjusted for sex and age (continuous)
- Results
Table 2: Based on your correction factor of 1.15 the third increase in the table should be 15%.
Response: Thank you for this comment. As suggested, we have corrected the figure in Table 2.
Table 3: except two estimated effects "Occupation (Nurse)" and "Number of days performing interventional fluoroscopy per month (>=20)" there is no relevant difference between the crude and the adjusted estimations - what is the derivation here?
Response: We adjusted for age and sex in the logistic regression analysis because occupational risk factors were our main interests. The sex distribution by occupational characteristics was not statistically significant in general, except for “occupation” (i.e., nurses are predominately female than those of other jobs) (please see the table below). This difference produced a significant change between the crude and adjusted estimates for the nurses. The ORs of the number of days performing interventional fluoroscopy per month (≥ 20 days) also slightly changed, as the reviewer pointed out, but the difference was not significant.
Table. Sex and age distribution by demographic and occupational characteristics
|
Characteristics |
Male |
Female |
||
|
N |
(%) |
N |
(%) |
|
|
Type of medical facility |
||||
|
General hospitals |
622 |
(73.1) |
229 |
(26.9) |
|
Others |
104 |
(75.4) |
34 |
(24.6) |
|
Occupation |
|
|
|
|
|
Technologists |
426 |
(90.8) |
43 |
(9.2) |
|
Doctors |
230 |
(83.9) |
44 |
(16.1) |
|
Nurses |
70 |
(28.5) |
176 |
(71.5) |
|
Specialty |
|
|
|
|
|
Radiology |
415 |
(74.4) |
143 |
(25.6) |
|
Cardiology |
146 |
(70.5) |
61 |
(29.5) |
|
Others |
165 |
(73.7) |
59 |
(26.3) |
|
Calendar year of beginning work |
||||
|
< 2010 |
258 |
(79.1) |
68 |
(20.9) |
|
2010–2014 |
189 |
(81.1) |
44 |
(18.9) |
|
≥ 2015 |
279 |
(64.9) |
151 |
(35.1) |
|
Duration of employment (years) |
||||
|
< 5 |
239 |
(64.8) |
130 |
(35.2) |
|
5–9 |
206 |
(74.9) |
69 |
(25.1) |
|
≥ 10 |
281 |
(81.4) |
64 |
(18.6) |
|
Number of days performing interventional fluoroscopy per month |
||||
|
< 10 |
90 |
(68.2) |
42 |
(31.8) |
|
10–19 |
121 |
(68.0) |
57 |
(32.0) |
|
≥ 20 |
427 |
(75.3) |
140 |
(24.7) |
|
Frequency of performing interventional fluoroscopy per week |
||||
|
< 10 |
177 |
(72.2) |
68 |
(27.8) |
|
10–29 |
208 |
(74.6) |
71 |
(25.4) |
|
≥ 30 |
249 |
(72.0) |
97 |
(28.0) |
|
Proportion of interventional fluoroscopy performed during their practice |
||||
|
100% |
233 |
(75.3) |
73 |
(24.7) |
|
75%–99% |
135 |
(67.8) |
64 |
(32.2) |
|
50%–74% |
103 |
(70.5) |
43 |
(29.5) |
|
25%–49% |
96 |
(76.8) |
29 |
(23.2) |
|
< 25% |
83 |
(74.1) |
29 |
(25.9) |
|
Wearing a lead apron |
|
|
||
|
Always |
682 |
(73.7) |
244 |
(26.3) |
|
Not always |
44 |
(69.8) |
19 |
(30.2) |
|
Wearing a thyroid shield |
|
|
|
|
|
Always |
602 |
(73.6) |
216 |
(26.4) |
|
Not always |
124 |
(72.5) |
47 |
(27.5) |
|
Wearing a goggle shield |
|
|
||
|
Always |
178 |
(82.8) |
37 |
(17.2) |
|
Not always |
548 |
(70.8) |
226 |
(29.2) |
|
Wearing a glove shield |
|
|
||
|
Ever |
115 |
(78.8) |
31 |
(21.2) |
|
Never |
611 |
(72.5) |
232 |
(27.5) |
Table 3: at least one relation sign is not correct - see: "Calendar year of beginning work" - the year 2015.
Response: Thank you for the clarification. We have corrected “2015” to “2014” in the Table 3.
- Discussion
- line 192: The underestimation is about 47% (not 90% as you report) if your derived corrected value is right.
Response: In this study, the percentage increase in radiation dose was calculated using the following equation: [100 × (adjusted dose − NDR dose)/NDR dose], as described in the Data analysis section. Because our purpose was to quantify the underestimation compared to the reported dose of NDR, the denominator should be the NDR dose, not the adjusted dose. Therefore, the level of underestimation was 89% [100 × (1.79 – 0.95)/0.95].
- Conclusion (line 263ff)
Under-reporting and under-estimation leads to biased risk estimates and should be avoided. Is there a viable way to take this into account?
Response: To avoid biased risk estimates in epidemiological studies, it is essential to increase the level of badge-wearing compliance among workers. If not, the alternative practical way is to conduct a sensitivity analysis excluding workers who had low reliable NDR doses (low compliance groups), as described in the Discussion.
To incorporate the reviewer’s comment, we revised the Conclusion as follows: “In summary, we provided quantitative information demonstrating that the NDR data may have underestimated the actual occupational radiation exposure level among medical radiation workers who perform fluoroscopically-guided interventional procedures in South Korea. Such an underestimation of occupational radiation doses may lead to biased risk estimates in epidemiological studies of radiation workers. Thus, considerable attention is required when interpreting and utilizing the NDR data, such as conducting a sensitivity analysis excluding workers who had low reliable NDR doses. Further studies are needed to evaluate the influence of the dose uncertainty induced by underestimation of the true doses due to the low dosimeter wearing compliance” (page 9, lines 287–296).
Reviewer 3 Report
This study assessed the extent to which individual dosimeter readings were underestimated in Korean interventional medical radiation workers. Based on responses to a survey about proportion of time wearing their dosimeters, the authors applied a correction factor to the doses reported in the National Dose Registry. The results are important because they highlight both the problem of inconsistent use and noncompliance of badge protocols among medical workers, and the issue of measurement error in epidemiologic studies of medical radiation workers (and the need for correct for this issue when possible). However, the study relied on self-reported badge use; this most likely have been overreported by participants. Also, some participants may have included in their reported badge use any time spent on activities unrelated to radiation exposure (see comment #1 below), resulting in an underreporting of badge use.
Specific comments are listed below, in the general order of importance:
1. I have some concerns about the use of the term “compliance” in this study, and whether this term accurately reflects what was measured in the survey question regarding proportion of badge use in this population (described in lines 91-93). For instance, is it possible that some participants reported less than 100% use of dosimeters during their practice because they have work-related tasks involving no radiation exposure? In this case, the worker would be fully compliant if he or she always wore a badge while working with radiation procedures but did not wear a badge during other work-related activities.
2. Introduction, 1st paragraph: Could the authors provide some more context about National Dose Registries? Could they list some countries that have them? This paragraph currently reads as if all, or at least, many countries maintain these databases.
3. How were badge readings under the minimum level of detection handled in this analysis?
4. Line 243-249: Is regular use of dosimeters defined the same way in all these other studies (wore badges 100% of the time)?
5. The authors should use consistent terminology when describing fluoroscopically-guided interventional procedures. In the paper, they use “fluoroscopy-guided interventional,” “fluoroscopic-guided interventional,” and “interventional fluoroscopy.”
6. Line 68, delete “were”
7. Line 92, change “enquired” to “asked”
8. Line 214: change to “Workers who specialize in radiology and cardiology may have more radiation safety education than those in other specialties because…”
9. Line 253: “nondifferential” should be “nondifferentially”
Author Response
This study assessed the extent to which individual dosimeter readings were underestimated in Korean interventional medical radiation workers. Based on responses to a survey about proportion of time wearing their dosimeters, the authors applied a correction factor to the doses reported in the National Dose Registry. The results are important because they highlight both the problem of inconsistent use and noncompliance of badge protocols among medical workers, and the issue of measurement error in epidemiologic studies of medical radiation workers (and the need for correct for this issue when possible). However, the study relied on self-reported badge use; this most likely have been overreported by participants. Also, some participants may have included in their reported badge use any time spent on activities unrelated to radiation exposure (see comment #1 below), resulting in an underreporting of badge use.
We thank the reviewer for the helpful comments. We have revised the manuscript in a point-by-point manner according to your comments, as described below.
Specific comments are listed below, in the general order of importance:
- I have some concerns about the use of the term “compliance” in this study, and whether this term accurately reflects what was measured in the survey question regarding proportion of badge use in this population (described in lines 91-93). For instance, is it possible that some participants reported less than 100% use of dosimeters during their practice because they have work-related tasks involving no radiation exposure? In this case, the worker would be fully compliant if he or she always wore a badge while working with radiation procedures but did not wear a badge during other work-related activities.
Response: We agree with the reviewer’s concern regarding the interpretation of compliance information in the questionnaire. The questionnaire related to dosimeter-wearing compliance represents the overall compliance level of radiation workers and may not capture detailed information about the frequency of wearing a badge by time period and work procedure. However, our study participants only included medical workers who performed fluoroscopically-guided interventional procedures and were primarily involved in radiation procedures. Further research with information on the time period and work procedures may help clarify the specific role of badge-wearing compliance.
To clarify these points, we have revised the sentence as follows: “The questionnaire related to dosimeter-wearing compliance represents the overall compliance level of radiation workers and may not capture detailed information about the frequency of wearing a badge by time period and specific work procedures.” However, the compliance level may be non-differentially misclassified among the participants, and information on self-reported working practices that involve radiation exposure has been generally reported as reliable among South Korean radiologic technologists [31]. Further research with information on time period and work procedures may help clarify the specific role of badge-wearing compliance (page 9, lines 271–279).
- Introduction, 1st paragraph: Could the authors provide some more context about National Dose Registries? Could they list some countries that have them? This paragraph currently reads as if all, or at least, many countries maintain these databases.
Response: The National Dose Registries contain the dose records of individuals monitored for occupational exposure to ionizing radiation. Many countries have national dose registries that contain dose records of individuals worldwide. Instead of listing the names of countries, we have added a sentence on the importance of NDR in the paragraph “Such registries are important parts of national occupational radiation protection programs in many countries” (page 1, line 32).
- How were badge readings under the minimum level of detection handled in this analysis?
Response: When the radiation dose was below the minimum detectable level (0.01 mSv), we applied an estimate of half of the detectable level (0.005 mSv) owing to a highly skewed distribution (Hornung RW; Reed LD. Estimation of average concentration in the presence of nondetectable values. Appl Occup Environ Hyg 1990; 5(1):46-51).
We have added the sentence and a reference accordingly in the Materials and Methods: “In cases where the dose was below the minimum detectable level, the dose was considered to be half of the detectable level due to a highly skewed distribution [11] (page 3, line 116).”
- Line 243-249: Is regular use of dosimeters defined the same way in all these other studies (wore badges 100% of the time)?
Response: Thank you for this comment. In this analysis, workers were categorized as regular and irregular badge-wearers (i.e., always vs. not always badge-wearing). However, we agree that the word ‘regular’ has a vague meaning and may have different definitions in different studies. To avoid this confusion, we have replaced the word “regular” with “always badge wearer” throughout the text.
- The authors should use consistent terminology when describing fluoroscopically guided interventional procedures. In the paper, they use “fluoroscopy-guided interventional,” “fluoroscopic-guided interventional,” and “interventional fluoroscopy.”
Response: We thank the reviewer for pointing this out. We have revised the text to maintain consistent terminology as “fluoroscopically-guided interventional procedures” based on international literature such as ICRP, NCRP, and other published papers.
- Line 68, delete “were”
Response: We have deleted this word from the sentence (page 2, line 74).
- Line 92, change “enquired” to “asked”
Response: We have corrected this word as suggested (page 3, line 98).
- Line 214: change to “Workers who specialize in radiology and cardiology may have more radiation safety education than those in other specialties because…”
Response: We have corrected this sentence as per the reviewer’s suggestion (page 8, line 232).
- Line 253: “nondifferential” should be “nondifferentially”
Response: We have corrected this word, as suggested (page 9, line 275).
We thank the reviewer for their helpful comments and hope that our revisions meet the requirements of the reviewers.